# Self-assembly of emissive supramolecular rosettes with increasing complexity using multitopic terpyridine ligands

Guang-Qiang Yin[1,2], Heng Wang[2], Xu-Qing Wang[1], Bo Song[2], Li-Jun Chen[1], Lei Wang[2], Xin-Qi Hao[3], Hai-Bo Yang[1] & Xiaopeng Li[2]

Coordination-driven self-assembly has emerged as a powerful bottom-up approach to construct various supramolecular architectures with increasing complexity and functionality. Tetraphenylethylene (TPE) has been incorporated into metallo-supramolecules to build luminescent materials based on aggregation-induced emission. We herein report three generations of ligands with full conjugation of TPE with 2,2′:6′,2″-terpyridine (TPY) to construct emissive materials. Due to the bulky size of TPY substituents, the intramolecular rotations of ligands are partially restricted even in dilute solution, thus leading to emission in both solution and aggregation states. Furthermore, TPE-TPY ligands are assembled with Cd (II) to introduce additional restriction of intramolecular rotation and immobilize fluorophores into rosette-like metallo-supramolecules ranging from generation 1–3 (**G1**–**G3**). More importantly, the fluorescent behavior of TPE-TPY ligands is preserved in these rosettes, which display tunable emissive properties with respect to different generations, particularly, pure white-light emission for **G2**.

[1] Shanghai Key Laboratory of Green Chemistry and Chemical Processes, Zhuang Chang Gong Institute, School of Chemistry and Molecular Engineering, East China Normal University, 3663 North Zhongshan Road, Shanghai, 200062, China. [2] Department of Chemistry, University of South Florida, Tampa, 33620, USA. [3] College of Chemistry and Molecular Engineering, Zhengzhou University, Zhengzhou, 450001, China. Guang-Qiang Yin and Heng Wang contributed equally to this work. Correspondence and requests for materials should be addressed to H.-B.Y. (email: hbyang@chem.ecnu.edu.cn) or to X.L. (email: xiaopengli1@usf.edu)

Among the diverse fields of supramolecular chemistry, coordination-driven self-assembly has emerged as a powerful bottom-up approach to construct various supramolecular architectures ranging from basic 2D macrocycles to large 3D cages with increasing complexity[1–10]. The breadth and depth of its scope is further evidenced by the variety of application benefiting from the precisely controlled size, shape, and composition of metallo-supramolecules[11–15]. With the aim to advance metallo-supramolecules with comparable sophistication as biological self-assembly, three strategies have been extensively approached, i.e., design new structures with increasing complexity, introducing functional moieties with broad diversity into well-defined supramolecular scaffolds, and host–guest encapsulation[16–19]. Within this field, the combination of chromophores with metallo-supramolecules as light-emitting materials attracted considerable attention because of their broad applications in light-emitting diodes, sensors, photoelectric devices, bioimaging, and so on[20–23].

Recently, tetraphenylethylene (TPE) as an archetypal fluorophore has been incorporated into metal–organic frameworks (MOFs)[24–26], covalent organic frameworks (COFs)[27–29], metallo-macrocycles[30–33], and metallo-cages[34–38] to construct luminescent materials based on aggregation-induced emission (AIE)[39], an effect caused by the restriction of intramolecular rotation (RIR)[40, 41].

In most cases of coordination system, RIR was achieved through anchoring TPE fluorophores to metal ions within rigid scaffolds in order to block the non-radiative path and activate the radiative decay; the TPE-containing ligands typically consist of small substitutes, e.g., carboxylate or pyridine for the coordination[21]. Therefore, RIR was mainly attributed to metal-coordination, by which the intramolecular motions are partially restricted.

In this study, we report the design and synthesis of three generations of AIE-active ligands with full conjugation of TPE with 2,2′:6′,2″-terpyridine (TPY)[42, 43], which is weakly luminescent ($\Phi_{em} = 3 \times 10^{-3}$) and becoming increasingly popular in metallo-supramolecular chemistry for the self-assembly of coordination polymers and discrete supramolecular architectures[44–47]. The larger conjugation and bulky size of the TPY substituents result in the intramolecular rotations partially restricted even in solution, and thus lead to emission in both solution and aggregation states. Furthermore, TPE-TPY ligands are assembled with Cd(II) through coordination to introduce additional RIR and immobilize fluorophores into metallo-macrocycles, or rosettes-like scaffolds. Using ditopic ligand, a mixture of macrocycles is obtained; in contrast, discrete double-layered hexameric and triple-layered heptameric rosettes with increasing structural complexity are assembled with tetratopic and hexatopic ligands, respectively, through multivalent interactions. More importantly,

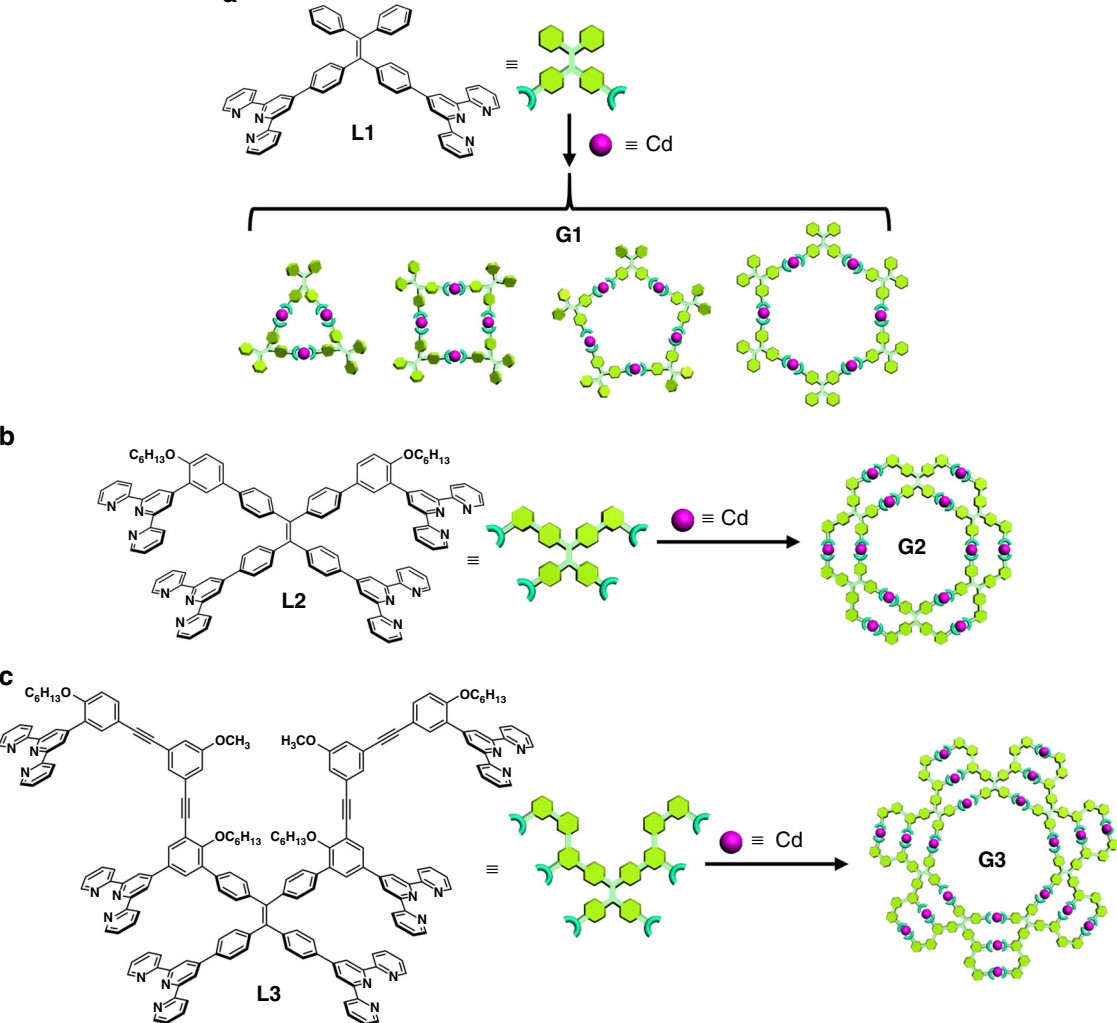

**Fig. 1** Self-assembly of supramolecular rosettes **G1**–**G3**. **a L1** assembled with Cd²⁺ to form a mixture of trimer, tetramer, pentamer, and hexamer macrocycles (**G1**); **b L2** assembled with Cd²⁺ to form a discrete hexamer (**G2**); **c L3** assembled with Cd²⁺ to form a discrete heptamer (**G3**)

the fluorescent behavior of TPE-TPY ligands is preserved in these rosettes, which display tunable emissive properties with respect to different generations.

## Results

### Synthesis and characterization of supramolecular rosettes G1−G3.
With the goal of increasing the RIR and structural complexity, **L1**, **L2**, and **L3** were synthesized by introducing multiple TPY groups onto TPE core through several steps of Suzuki or/and Sonogashira couplings in decent yields as shown in Supplementary Figures 1-3. All the ligands were fully characterized by nuclear magnetic resonance (NMR), including $^1$H, $^{13}$C, two-dimensional correlation spectroscopy (2D-COSY), nuclear Overhauser effect spectroscopy (2D-NOESY), rotating frame nuclear Overhauser effect spectroscopy (2D-ROESY), high-resolution electrospray ionization time-of-flight (ESI-TOF) and matrix-assisted laser desorption/ionization time-of-flight (MALDI-TOF) mass spectrometry. They were assembled with $Cd(NO_3)_2$ in exact stoichiometric ratios in CHCl$_3$/MeOH mixed solvent to form three generations of supramolecular rosettes, **G1−G3**, without any separation in high yields (Fig. 1). Note that a mixture of macrocycles was obtained in the self-assembly of **L1**,

instead of discrete hexamer (**G2**) and heptamer (**G3**) assembled by **L2** and **L3**, respectively.

The formation of the macrocyclic structures was first documented by NMR spectroscopy (Fig. 2 and Supplementary Fig. 38). Compared with the sharp $^1$H-NMR signals of the ligands, the spectra of the supramolecules, **G1−G3**, display remarkable broaden peaks of all protons, due to their much slower tumbling motion on the NMR time scale[4]. These peaks of **G2** (Supplementary Fig. 59) and **G3** (Supplementary Fig. 60) were getting much sharper with increasing temperature from 293 to 333 K. For instance, in the spectra of **L2** and **G2** (Fig. 2a), signals of 3′, 5′ and a3′, a5′ protons of **G2** shifted downfield (ca. 0.3 p.p.m.) compared with those signals of **L2**; while peaks assigned to 6, 6″ and a6, a6″ protons of TPY shifted significantly upfield (ca. 0.7 p.p.m.) due to the electron shielding effect[47]. These characteristic shifts are caused by TPY−Cd(II) coordination and consistent with the previous reports[43]. The spectra of **G3** (Fig. 2b) and **G1** (Supplementary Fig. 38) showed the similar variation of the signal locations. All NMR resonances of ligands and supramolecular rosettes were unequivocally assigned by 2D-COSY, and NOESY (Supplementary Figs. 15–60), indicating that a set of macrocycles were assembled rather than random linear supramolecular polymers.

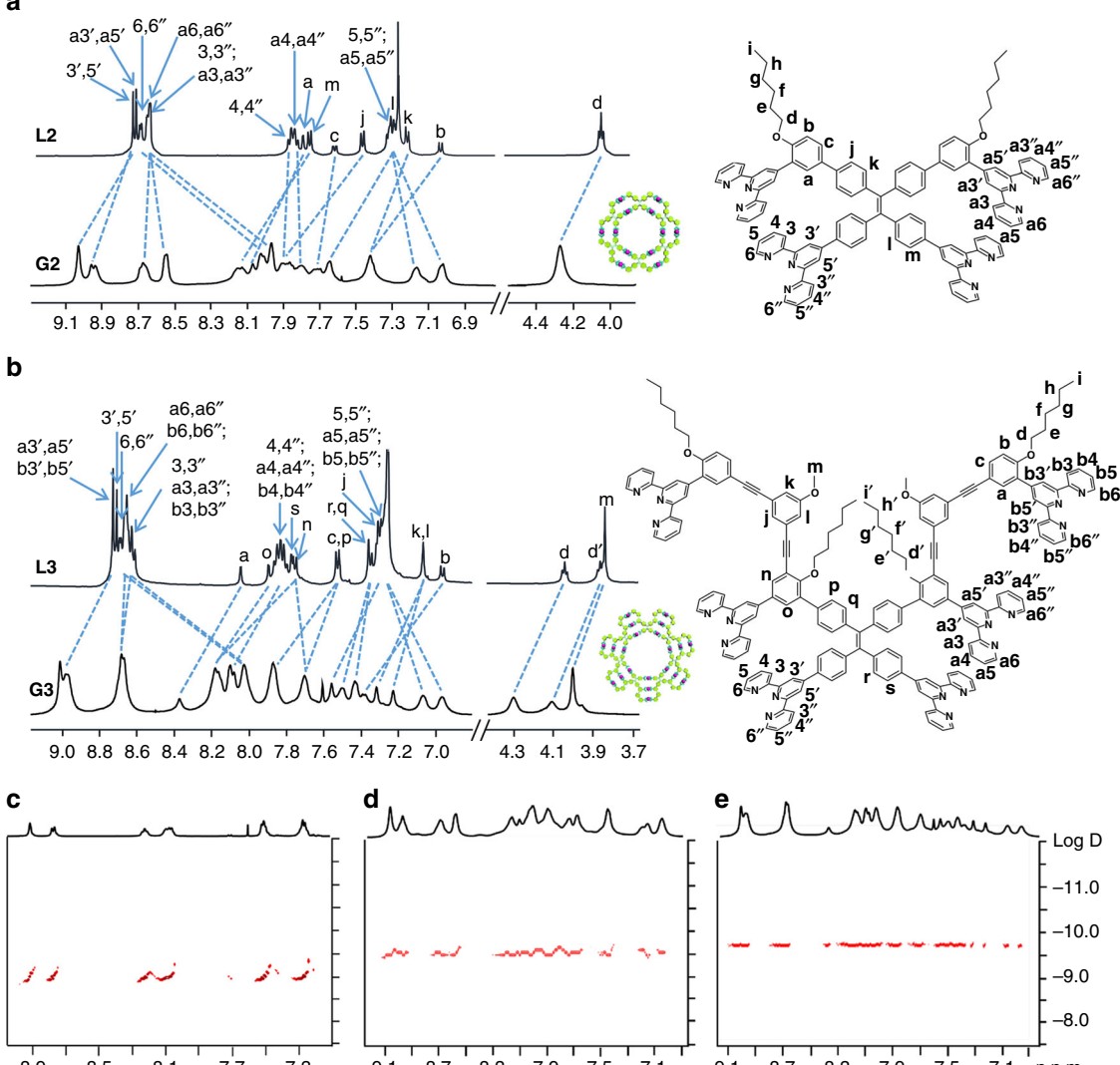

**Fig. 2** $^1$H-NMR spectra. **a** L2 and G2; **b** L3 and G3; **c** DOSY of G1; **d** DOSY of G2; **e** DOSY of G3 (500 MHz, 300 K, CDCl$_3$ for ligands and CD$_3$CN for supramolecules)

The molecular compositions of the supramolecular rosettes were further obtained with the aid of ESI-TOF MS and traveling-wave ion mobility mass spectrometry (TWIM-MS)[48]. The ESI-TOF MS of **G1** shows that it is a mixture of trimer, tetramer, pentamer, and hexamer (Supplementary Figs. 5 and 8) in the self-assembly of **L1** with Cd(II). It is consistent with the previous reports that the angle between two TPY groups of the ditopic ligands are flexible to form various rings[47]. Furthermore, the components of **G1** could be varied by changing the concentration (Supplementary Fig. 11). In contrast, **G2**, shows one series of peaks with continuous charge states from 10+ to 19+ in ESI-MS (Fig. 3a), corresponding to successively losing of $PF_6^-$ counterions. As expected, the averaged measured molecular mass of **G2** is 14498 Da, corresponding to a double-layered hexameric rosette with the formula of $[(C_{110}H_{88}O_2N_{12})_6Cd_{12}(PF_6)_{24}]$. The experimental isotope pattern of each charge state agrees very well with the theoretical distribution (Supplementary Fig. 9). The TWIM-MS of **G2** (Fig. 3b) displays a single set of signals with narrow drift time distribution at each charge state, suggesting the formation of a rigid and discrete assembly. Therefore, multivalent interactions introduced by the tetratopic ligand resulted in high density of coordination sites (DOCS)[47, 49] and provided more geometric constraints to prevent self-assembly from forming multiple entities and to reach the most thermodynamically preferable structure.

With increasing structural complexity, **G3** also displays one dominant set of peaks (11+ to 21+) in ESI-MS (Fig. 3c). Surprisingly, the averaged measured molecular mass of **G3** after deconvolution is 27,599 Da, which perfectly matches with a formula $[(C_{186}H_{150}O_6N_{18})_7Cd_{21}(PF_6)_{42}]$ corresponding to a triple-layered heptameric rosette structure instead of hexamer. It should be noted that the minor signals ($F^{n+}$) were assigned carefully as the fragments of **G3** due to high voltage applied in ESI source to improve the ionization efficiency of large assemblies (Supplementary Fig. 6). The ESI-MS of **G3** with different ionization voltages provided further evidence for the formation

of these fragments (Supplementary Fig. 7). The TWIM-MS spectrum of **G3** (Fig. 3d) was also recorded and presented a narrowly distributed band of signals. The experimental and calculated isotope patterns of each charge state of **G3** are also summarized in Supplementary Fig. 10. In the well-documented metallo-macrocycle self-assembly[16], it is very rare to construct discrete triple-layered heptameric macrocycles through direct self-assembly without separation in high yield. We speculated that the formation of **G3** with heptamer composition instead of hexamer could be attributed to the geometry feature of **L3**. According to the molecular simulation of hexamer and heptamer assembled by **L3**, the structure of hexamer (Supplementary Fig. 14c) exhibits higher torsion energy (567.91 kcal mol$^{-1}$) than that (499.27 kcal mol$^{-1}$) of heptamer. Therefore, the self-assembly preferred the formation of heptamer rather than hexamer.

The sizes and shapes of the supramolecules can be validated by further analyzing the collision cross sections (CCSs) obtained from TWIM-MS[47, 49]. Experimental CCSs values of **G2** and **G3** on each charge state are summarized in Supplementary Table 1. The averaged measured CCSs value of **G2** is 2041 Å$^2$, which is smaller than that of the higher generation assembly, **G3** (3821 Å$^2$). Through molecular dynamics simulation (Supplementary Figs. 12–13), the averaged calculated CCSs from 70 candidate structures of **G2** and **G3** are 2065 and 3981 Å$^2$, respectively. The good agreement of calculated and experimental results further supports the rosettes structures of the assemblies.

The 2D diffusion-ordered NMR spectroscopy (DOSY) experiments provide more structural information on the size and purity of **G1−G3** (Fig. 2c−e). Strongly consistent with the ESI-MS results, DOSY spectrum of **G1** shows at least three well-split signal bands, indicating the mixed cyclic structures in **G1**. By contrast, narrow dispersed band of signals was displayed in the DOSY spectra of **G2** and **G3**, suggesting that no other structures existed in the self-assembly. The measured diffusion coefficients ($D$) of **G1−G3** are gradually decreased from $9.07 \times 10^{-10}$ m$^2$ s$^{-1}$ (the average value of **G1** mixtures) to $2.51 \times 10^{-10}$ m$^2$ s$^{-1}$ (**G2**)

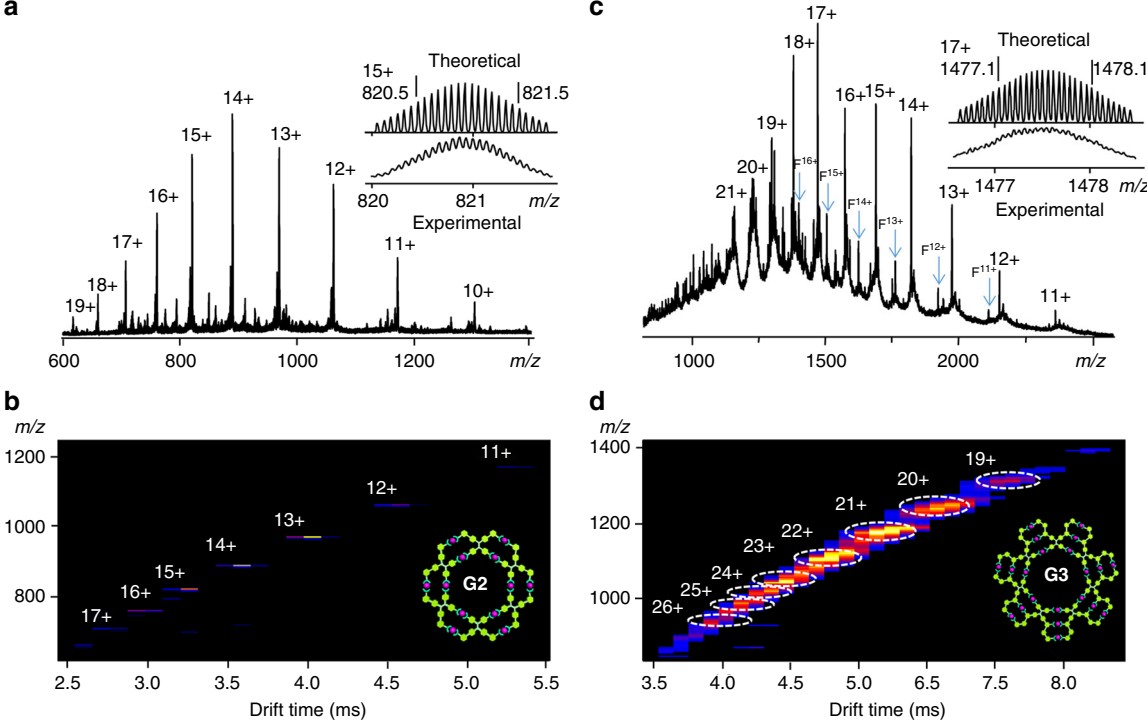

**Fig. 3** ESI/TWIM-MS spectrum. **a** ESI-MS and **b** TWIM-MS plots of **G2**; **c** ESI-MS and **d** TWIM-MS plots of **G3**. The peaks of $F^{n+}$ denote fragments $[\mathbf{G3}-\mathbf{L3}-nPF_6^-]^{n+}$

and $1.78 \times 10^{-10} \, m^2 \, s^{-1}$ (**G3**). The decreasing $D$-values are corresponding to the increasing size of these supramolecules, consistent with the results obtained from TWIM-MS and molecular simulation.

Transmission electron microscopy (TEM) was utilized to image the giant 2D supramolecular rosettes, **G2** and **G3**, in order to further confirm their shapes and sizes. In TEM image (Fig. 4b, f), individual circular patterns were clearly observed for both **G2** and **G3**. The size information obtained from TEM are comparable to the theoretical diameter of 6.3 and 8.6 nm for **G2** and **G3** (Fig. 4a, e), respectively. We also collected atomic force microscopy (AFM) images of **G2** and **G3** on mica surface (Fig. 4i–k,l–n, and the statistical height histograms of the particles were summarized in Supplementary Figs. 99–100). Both rosette

shaped supramolecules displayed high density of individual dots, among which major part of the dots showed comparable measured height ca. 2.0 nm. All attempts to grow X-ray-quality single crystals of **G2** and **G3** have so far proven unsuccessful. Nevertheless, we obtained fiber-like nanostructures for both **G2** and **G3** (packing cartoon shown as Fig. 4c, g). TEM imaging showed the formation of tubular structures through the stacking of individual supramolecular rosettes (Fig. 4d, h, and Supplementary Figs. 101–102). The diameters of nanotubes are consistent with those of individual supramolecule by molecular modeling given the contribution from alkyl chains.

**Emission and AIE effects of L1−L3.** After structural characterization of organic ligands and supramolecules, we conducted detailed photo-property studies. Considering the solubility of **L1**

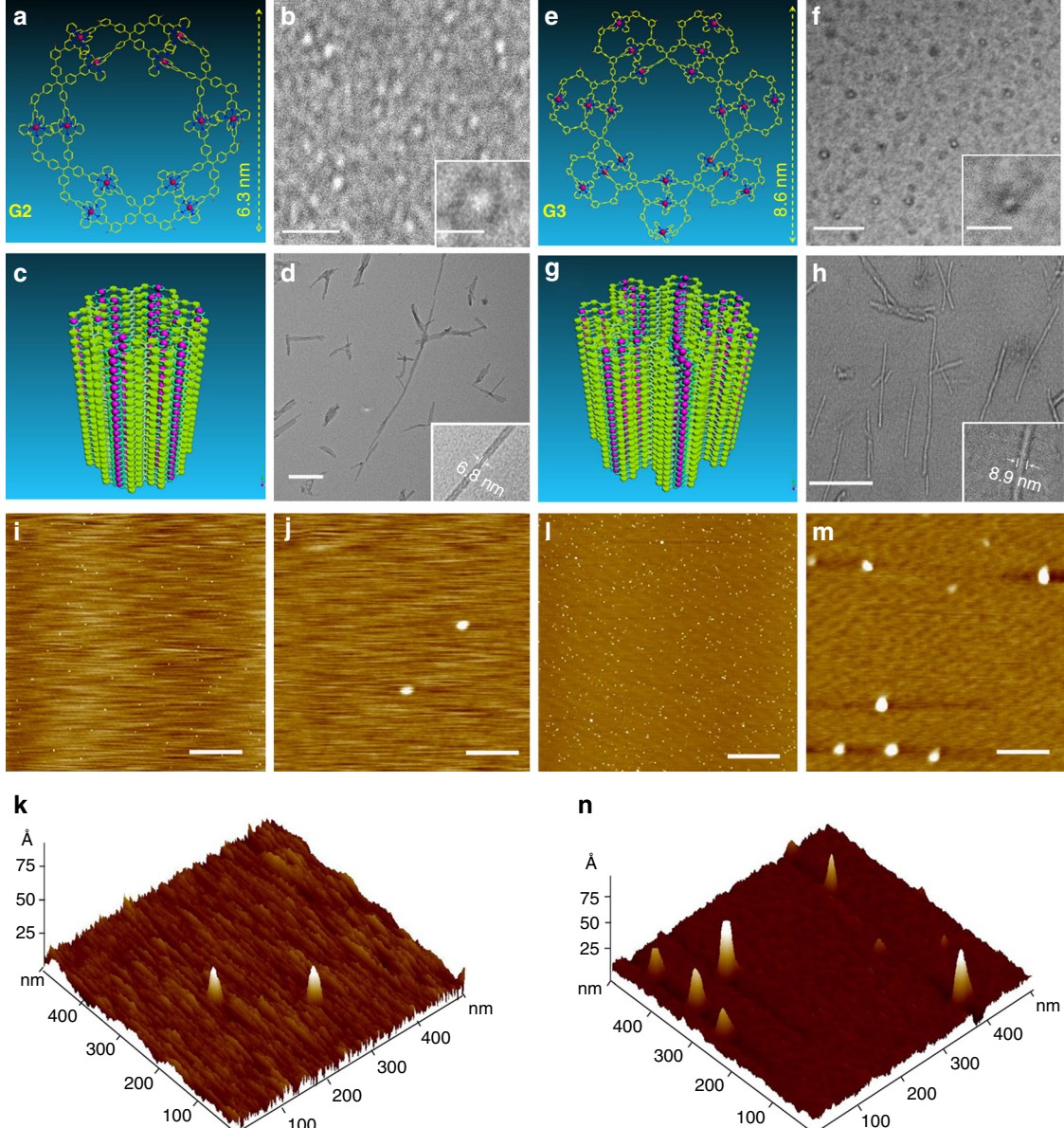

**Fig. 4** TEM and AFM images of **G2** and **G3**. Representative energy-minimized structure from molecular modeling of **a** **G2** and **e** **G3** (alkyl chains are omitted for clarity); TEM images of single molecular **b** **G2** (scale bar, 30 nm and 10 nm for zoom-in image) and **f** **G3** (scale bar, 100 nm and 30 nm for zoom-in image); proposed stacking structure of **c** **G2** and **g** **G3**; TEM images of nanotubes assembled by **d** **G2** (scale bar, 200 nm) and **h** **G3** (scale bar, 200 nm) 0.5 mg mL$^{-1}$ in acetonitrile solution under isopropyl ether vapor; AFM images of **i**, **j** **G2** (scale bar, 2 μm and 100 nm, respectively) and **l**, **m** **G3** (scale bar, 2 μm and 100 nm, respectively); 3D AFM images of **k** **G2** and **n** **G3**

−L3, we eventually chose $CH_2Cl_2$ and methanol as good and poor solvent, respectively, to perform the studies (Supplementary Fig. 61, absorption spectra; Supplementary Figs. 68–70, emission spectra). All these three ligands are non/weak-luminescent in $CH_2Cl_2$, but showed apparent AIE effect by gradually increasing the volume fraction of the poor solvent, methanol. It is worth noting that **L2** and **L3** displayed stronger emission in pure $CH_2Cl_2$ than **L1** due to the RIR caused by multiple bulky TPY groups. Also, **L2** and **L3** showed AIE effect with high efficiency and quantum yields ($\Phi_F$) at 65.5% and 55.6%, respectively (determined in 90% methanol fraction, Supplementary Figs. 69 –70). By contrast, **L1** displayed much weaker AIE effect in various methanol fraction mixture ($\Phi_F < 5\%$). In addition, variation of the topology of the ligands tuned the maximum emission wavelength from 425 nm (**L1**) to 580 nm (**L2**, dual emission, another peak centered at 440 nm), and 475 nm (**L3**), under the aggregation state (90% methanol fraction). The distinguished dual emission bands of **L2** are attributable to the local excited (LE) state of TPE (shorter wavelength), as well as the intramolecular charge transfer (ICT) from electron donor (TPE) to the electron acceptor (TPY) moieties (longer

wavelength)[50]. Under the solid state (Supplementary Fig. 79), all the three ligands emit long wavelength light centered at around 520 nm, ascribed to ICT states.

**Emission and AIE effects of supramolecular rosettes G1−G3.** We reasoned that through further RIR by coordination, the assemblies with rosettes structures would exhibit high emission efficiency not only in aggregation, but also in solution state. The absorption spectra of **G1−G3** were summarized in SI (both solution and aggregation states, Supplementary Figs. 62–63). The emission spectra of the supramolecules, **G1−G3**, in acetonitrile as the good solvent were recorded in Figs. 5, 6 and Supplementary Figs. 71–72. In solution, as expected, **G2** and **G3** displayed remarkable increment in $\Phi_F$ values (2- and 6-fold, respectively) compared to their corresponding ligands, **L2** and **L3**. It is bene-fited from the RIR effect caused by the rigid supramolecular scaffold, in which, the rotation around the TPE groups is further restricted. By comparison, **G1** in solution remained weak emis-sion as its ligand, due to its freely rotating phenyl groups outside the metallo-macrocycle rings.

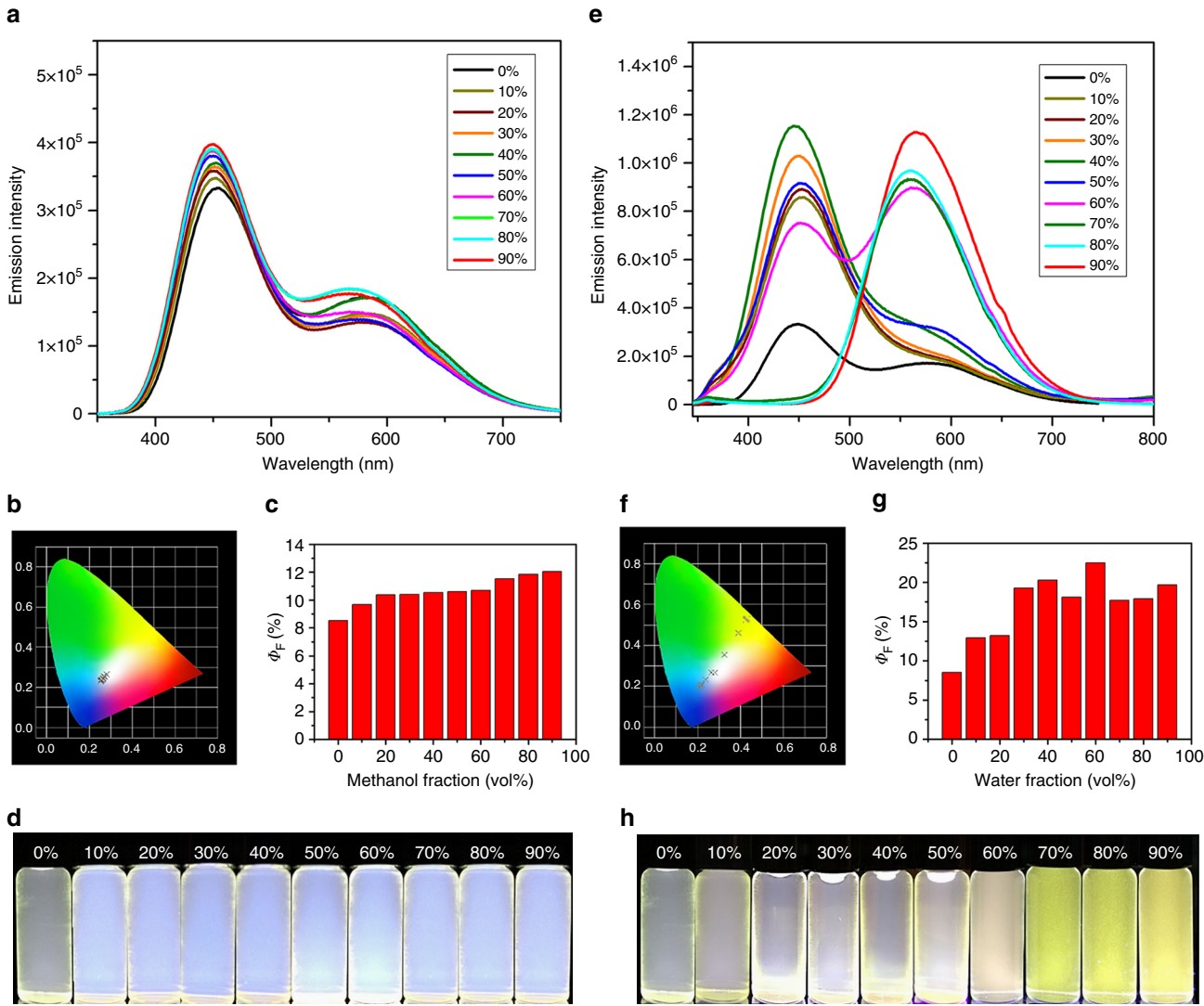

**Fig. 5** AIE of **G2**. **a** Fluorescence spectra ($\lambda_{ex} = 320$ nm, $c = 1.0$ μM), **b** CIE 1931 chromaticity diagram, (the crosses signify the luminescent color coordinates), **c** quantum yields, and **d** photographs of **G2** in $CH_3CN$/methanol with various methanol fractions; **e** fluorescence spectra ($\lambda_{ex} = 320$ nm, $c = 1.0$ μM), **f** CIE 1931 chromaticity diagram, **g** quantum yields, and **h** photographs of **G2** in $CH_3CN$/water with various water fractions. **G2** samples were excitation at 365 nm on 298 K ($c = 1.0$ μM)

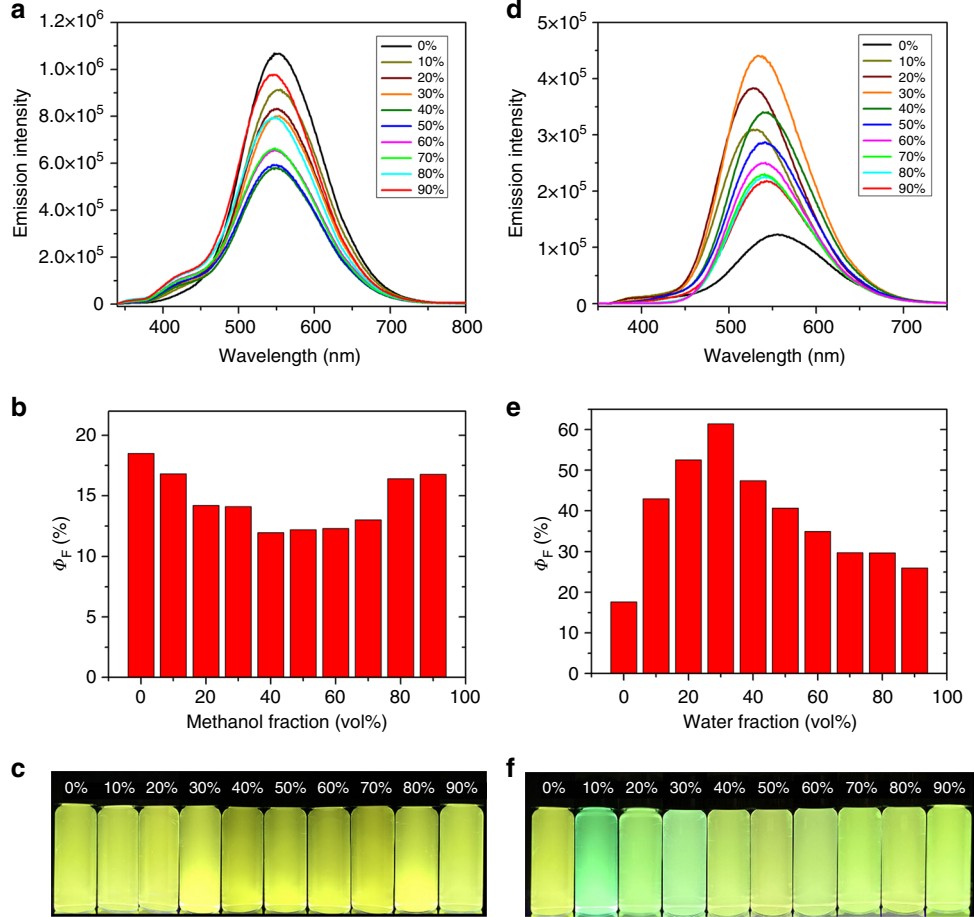

**Fig. 6** AIE of **G3**. **a** Fluorescence spectra ($\lambda_{ex}$ = 320 nm, $c$ = 1.0 μM), **b** quantum yields, and **c** photographs of **G3** in CH$_3$CN/methanol with various methanol fractions; **d** fluorescence spectra ($\lambda_{ex}$ = 320 nm, $c$ = 1.0 μM), **e** quantum yields, and **f** photographs of **G3** in CH$_3$CN/water with various water fractions. **G3** samples were excitation at 365 nm on 298 K ($c$ = 1.0 μM)

In sharp contrast with **L1**, macrocycles mixture, **G1**, shows gradually enhanced $\Phi_F$ with gradually increasing methanol fraction, i.e., from 0.3% with 0% methanol to 6.7% with 90% methanol. The emission band of **G1** (centered at ca. 530 nm) under aggregated state (90% methanol) exhibits 100 nm red shifted than its ligand, ascribed to ICT emission. In order to further investigating the AIE properties of **G1**, a poorer solvent, water, was used instead of methanol, to facilitate the aggregation of **G1**. By increasing the water content, $\Phi_F$ value was increased rapidly from around 10 to 70% (Supplementary Fig. 72), an extraordinary enhancement compared with the value tested in acetonitrile/methanol mixture, indicating more intense aggregation of **G1** in water environment.

More intriguingly, the optical feature of **G2** in solution and aggregation state showed rare room temperature white-light emission[38, 51]. The emission spectrum of **G2** in pure acetonitrile (Fig. 5a) displayed two main bands centered at 450 nm and 580 nm, ascribed to the LE and ICT states, respectively. Compared with **L2**, **G2** displayed a ca. 30 nm redshift, as well as broader bands in emission curve, which covered almost the entire visible spectral region (~400−700 nm). By increasing the fraction of methanol, intensity of the ICT band slightly increased, because the polar solvent facilitates the ICT process[50]. Changing the poor solvent to a higher polar one, water, further enhanced the intensity of the ICT band and suppresses the LE band (Fig. 5e), and consequently changed the emission color. In the case of 60% water fraction, **G2** emitted pure white light with coordination (0.325, 0.355) (Fig. 5f) in 1931 Commission Internationale de

L'Eclairage (CIE) chromaticity diagram, extremely close to the value of theoretical white light (0.333, 0.333)[38, 51]. In the case of 90% water fraction, only ICT band was observed, and the solution emitted yellow light instead of white light. **G2**, as a discrete rosette-like supramolecule, could be an outstanding candidate for white-light emission and color tunable optoelectronic materials.

The $\Phi_F$ values of **G2** in both acetonitrile/methanol (8−12% shown in Fig. 5c) and acetonitrile/water systems (8−20% shown in Fig. 5g) increased with the increment of the fraction of poor solvents, corresponding to AIE effect. The slow enhancement in quantum yields is perhaps partially brought about by aggregation-caused quench (ACQ) effect[51]. The ACQ effect probably comes from the face-to-face stacking structure of **G2** within the aggregates, due to the large and rigid 2D structures of **G2** assembled by all π conjugated moieties. Consequently, $\Phi_F$ values remained at a moderate level because of the opposite effects on emission intensity.

The emission behavior of **G3** was also investigated in both acetonitrile/methanol and acetonitrile/water solvents. The spectra show ICT emission peaks centered at ca. 550 nm with green−yellow color (Fig. 6c, f). The $\Phi_F$ values of **G3** in acetonitrile/water system first increased and then decreased by gradually increasing the fraction of water, with maximum $\Phi_F$ value as 60% (30% water fraction, Fig. 6e). It is also attributed to the opposite effects from AIE of TPE groups and ACQ of π stacked moieties as described in the case of **G2**. In acetonitrile/methanol system, $\Phi_F$ values slightly decreased, and exhibited a minimum point at 40% methanol ($\Phi_F$ = 12%, Fig. 6b), and then

increased again by adding more amount of methanol. It indicates that in acetonitrile/methanol system, the starting and saturated point of ACQ effect appeared at lower fraction of poor solvent than AIE effect; the competition between ACQ and AIE; however, was reversed in the acetonitrile/water system.

2D-fluorescence measurements of **G1−G3** (Supplementary Figs. 73−78) are consistent with 1D-fluorescence results. No other light-emitting species was observed. The 2D-fluorescence results of supramolecular rosettes show the independence of emission from excitation. The ICT bands (emission maxima and peak shape) of **G1−G3** are strongly sensitive to the solvent polarity (Supplementary Figs. 84−86). It is consistent with the reported CT emissions[50]. The intensity of LE emission of **G1** (Supplementary Fig. 95) and **G3** (Fig. 6a) is relatively low, due to the flexibility of the TPE backbone (**G1**) or suppressed by strong ICT emission (**G3**). Furthermore, as shown in Supplementary Figs. 92-94, the overlapped emission spectra and the absorption spectra of **G1−G3** are prone to the energy transfer (ET) process[52]. In addition, the strongest ICT effect of **G3** among three rosettes leads to the longer wavelength emission dominant. Emission spectra of **G1−G3** under solid state were also recorded in Supplementary Fig. 80, in which all of the assembly powders show single and broad peaks centered at around 500 nm. All these peaks are assigned to ICT process. Compared with the corresponding spectra recorded in the aggregation state, for instance, 90% water fraction, all these peaks show around 50 nm blueshift, because the polar solvent, water, facilitates charge separation of ICT process. Lifetimes of the supramolecules in both air saturated and degassed solution were determined with all

values in ns scale, suggesting a typical fluorescent emission (Supplementary Figs. 88−91). In order to further investigate fluorescence process, temperature dependent fluorescence spectra of **G1−G3** (Supplementary Figs. 81−83, from −46 to 50 °C) and concentration dependent spectra of **G1** (Supplementary Figs. 87) were also recorded. In addition, the association constants of model compound **16** (Supplementary Fig. 4) binding with Cd (NO$_3$)$_2$ were measured in different solvents by titration experiments (Supplementary Figs. 64−67). For example, the overall association constant in CHCl$_3$/MeOH (v/v, 1/2) is $4.17 \times 10^{10}$ M$^{-2}$.

To further investigate the emission properties of **G1−G3**, we studied their aggregation behaviors by TEM and dynamic light scattering (DLS). **G2**, for instance, aggregated into nanosphere particles in acetonitrile/methanol, revealed by TEM images (Fig. 7c−j), and the size of the particles were increased along with the increment of methanol fraction. DLS results also show that the average hydrodynamic diameters ($D_h$) of the nanospheres increased from 16 (20% methanol) to 24 nm (40% methanol), 37 nm (60% methanol), and 78 nm (80% methanol), with narrow distribution (Fig. 7a). Compared with the results in acetonitrile/methanol mixtures, **G2** aggregated into much larger nanospheres in 20% and 40% water contents (108 nm and 285 nm, respectively, Fig. 7b, k−n), and the nanospheres merged into necklace-like aggregates in higher water contents(Figs. 7o−r), in which the aggregates were too large for DLS measurements. These results well agree with the higher AIE effect observed in acetonitrile/water than in acetonitrile/methanol mixtures. Similar

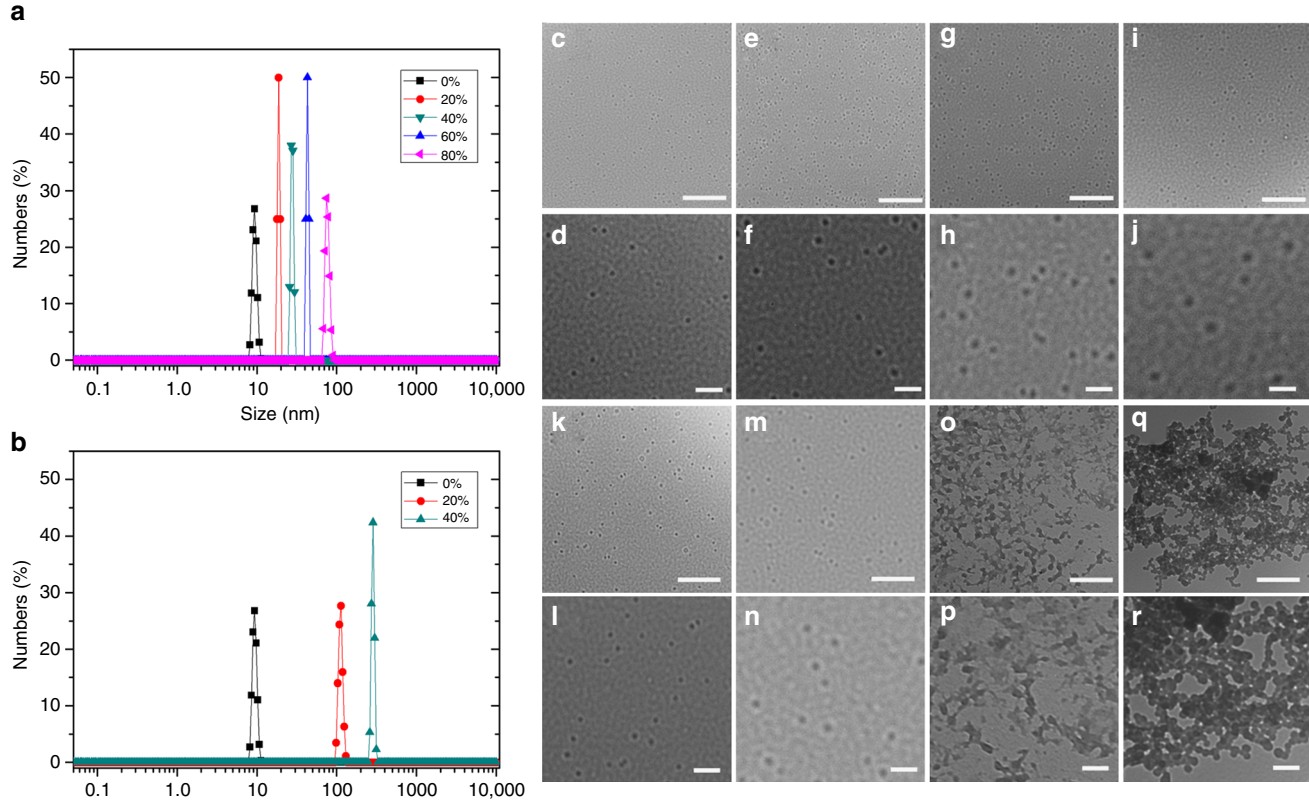

**Fig. 7** DLS data and TEM images of **G2** aggregates. Size distribution of **G2** in **a** acetonitrile/methanol, and **b** acetonitrile/water mixtures by DLS (the percentages in the graphs are the poor solvent contents); TEM images of the aggregates of **G2** formed in acetonitrile/methanol mixtures containing **c**, **d** 20%, **e**, **f** 40%, **g**, **h** 60%, and **i**, **j** 80% methanol (scale bar 500 nm for the upper images and 100 nm for the lower images, respectively), and aggregates of **G2** formed in acetonitrile/water mixtures containing **k**, **l** 20% (scale bar, 2 μm and 500 nm, respectively), **m**, **n** 40% (scale bar, 2 μm and 500 nm, respectively), **o**, **p** 60% (scale bar, 1 μm and 500 nm, respectively), and **q**, **r** 80% water (scale bar, 1 μm and 500 nm, respectively)

aggregation behaviors were observed in **G1** and **G3** systems (Supplementary Figs. 96–98 and 103–106).

## Discussion

In summary, we report the design and synthesis of three generations of AIE-active ligands with full conjugation of TPE with TPY. Instead of forming a mixture of macrocycles by ditopic ligands, such multitopic building blocks provided more geometric constraints in the self-assembly through multivalent interactions to prevent the formation of multiple entities and to reach the most thermodynamically favorable structures, i.e., double-layered hexameric and triple-layered heptameric rosettes-like scaffolds in a precisely controlled manner. Along with the increasing structural complexity, the multiple bulky TPY groups and multivalent interactions by coordination introduce enhanced RIR to immobilize TPE fluorophores into metallo-supramolecular architectures. In addition to preserving the emissive property of individual building block, such assembled supramolecular rosettes exhibit unique properties that are not displayed by their individual components. Remarkably, **G2** exhibited highly pure white-light emission property under wide range of good/poor solvents ratios. As such, this study may provide an alternative strategy in the seeking of novel light-emitting materials.

**Data availability**. The data that support the findings of this study are available from the authors on reasonable request, see author contributions for specific data sets.

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

## Acknowledgements

This research was supported by the National Science Foundation (CHE-1506722 to X.L.) and National Natural Science Foundation of China (Nos. 21625202, 21672070, and 91427304 to H.-B.Y.; No. 1528201 to X.L. and X.-Q. H.) G.-Q.Y. acknowledges the visiting scholarship from East China Normal University. We also thank Dr. Anjun Qin from the South China University of Technology for his thoughtful suggestions.

## Author contributions

X.L. conceived and designed the experiments. G.-Q.Y., H.W., L.W., and B.S. completed the synthesis. G.-Q.Y., H.W., X.-Q.W., B.S., L.-J.C. and X.-Q.H. conducted the characterization. G.-Q.Y., H.W., H.-B.Y. and X.L., analysed the data and wrote the manuscript. All the authors discussed the results and commented on and proofread the manuscript.

## Additional information

**Competing interests:** The authors declare no competing financial interests.

