## [Peer Review File · Nature Communications]

Reviewers' comments:

Reviewer #1 (Remarks to the Author):

Yin et al. described the synthesis, characterization and luminescence properties of a class of molecular rosettes based on the chelates between terpyridine-substituted tetraphenylethene (TPE-TPY) ligands and Cd (II). They found that by modulation of the number and position of TPY substitution on TPE, three "generations" of molecular rosettes structures could be formed in solution, as evidenced by NMR, MS, TEM and AFM characterization. The authors then proceeded to report the AIE behaviors of these molecular assemblies via fluorescence spectroscopy and TEM as well.

The content in the manuscript is well clarified and experiments carefully performed, using multiple corroborative methods to be extra thorough. There is certain novelty in the work although potential impact could be limited due to the characterization conditions (mix organic solvents). Furthermore, there are certain issues that are not clear but critical for the publication of the manuscript.

1) the total species in the chelate solution for G1-G3. The authors did not mention anywhere in the manuscript the association constants in various solvents and how might changing solvent composition affect these constants. I almost have the impression that these rosettes are the sole species in solutions, is that true? But I don't think the authors specified in the manuscript. If this is the case, the authors should provide a simple 2D-fluorescence measurement that indicates independence of emission from excitation for G1-G3 and how solvent composition influences the results. Also, a temperature dependent experiment may also help provided that the rosettes dissociate at higher temperatures.

2) the interpretation of certain fluorescence results, for instance, the assignment of the shoulder emission at longer wavelength throughout the samples. The authors attribute these peaks to charge-transfer fluorescence while providing no evidence to support it. For example, charge-transfer absorption and fluorescence are typically very sensitive solvent polarity and media rigidity. This should be very easy to prove with just a bit more effort in experiment. Also, the authors explained the enhancement of G1-G3 fluorescence in comparison to L1-L3 in good solvent as inhibition of molecular motions. This is hardly convincing given that the enhancement is not as dramatic. Metal coordination can have a tremendous effect in the electronic transitions of the ligands. I suggest that the author show hard evidence for all the explanations they list in the fluorescence properties section as they did in the characterization part at the beginning of the manuscript, which is nice and clear.

Some minor issues:

1) the description of the figure legends are not adequately informative (such as the lack of concentration information) and the readers have to fish out some numbers in the main text;

2) P10 L171 "rear"  "rare"?

3) what is the potential use of white emission in solution since the authors seem to emphasize the concept a lot?

4) I am just curious what the purpose is for choosing Cd (II) over other metals such as Zn(II)?

5) Measured lifetimes in the ns range do not necessarily mean fluorescence for G1-G3. If the quantum yield is only half a percent in air, then the intrinsic lifetimes are in the sub micron scale, which could involve metal in the triplet excited state.

Reviewer #2 (Remarks to the Author):

The manuscript describes the synthesis and characterization of three novel Cd(II)-tpy-based metallomacrocycles containing an AIE fluorophore, TPE, as well as the emissive properties of ligands and complexes in solution and in the solid state. The unique rosette-like supramolecular structures were produced upon complexation with Cd(II) ions. Particularly, the G2 hexamer showed white light emission. The molecular design and photophysical properties are very interesting. However, the following issues concerning the molecular characterization and the mechanisms responsible for the emission should be emphasized and elaborated in more detail to strengthen the conclusions.

- 1) Since the structures of ligands L1-L3 are very complicated, the detailed ¹H NMR assignments should be done more carefully. For example, the two sets of tpy signals should be distinguishable by NOE experiments. In addition, to ensure high purity of ligands, it is suggested the authors provide the whole spectrum of L1-L3 instead of the isotope patterns, or their elemental analysis.
- 2) The complexation of L1 with Cd(II) ions gave rise to a dynamic library including four macrocycles, which were observed by ESI-MS. Is it possible to have smaller or bigger rings, such as a dimer or a heptamer, in this series? Are those structures concentration-dependent? If yes, the population of the dynamic library should be very different at various concentrations. Therefore, how do the concentration and/or molecular sizes affect the emissive properties?
- 3) In the assembly of L2 with Cd(II), the resultant complex was assigned to a hexameric structure mainly relied on the ESI-MS evidence. However, there are some minor signals shown in Figure 3a. Can those minor signals be identified? Are they coming from other possible structures? Moreover, the broad ¹H NMR peaks were attributed to slow tumbling motions by the authors. Does the variant-temperature NMR experiment help understand this phenomenon? In the NMR spectrum, some signals are not identified properly, including two small humps at 8.8 and 8.4 ppm and a number of peaks in the range from 2.4 to 4 ppm. Are they generated by other assembled structures? If so, "quantitative yield" would be not true.
- 4) In the case of G3, again the broad ¹H NMR peaks and only a few signals in the ¹³C NMR spectrum were observed. There are even more noisy signals shown in the ESI-MS spectrum. On the basis of those data, the structure of the uncommon heptameric macrocycle cannot be established unambiguously. At least, the whole ESI-MS spectrum of G3 should be provided and carefully assigned to exclude the possibility of forming other species.
- 5) Although the tube-like morphologies were observed in the TEM images, the nanostructures cannot be directly correlated to the proposed molecular packing without any electron or x-ray diffraction evidence.
- 6) In the manuscript, the emission at shorter wavelength was attributed to the LE state of TPE and the one at longer wavelength was assigned to ICT from TPE to tpy units. The authors should provide experimental evidence to say so. In addition, since the TPE in G1 has higher flexibility, it would be expected to observe the emission at around 430 nm from TPE at higher solvent polarity due to AIE, but the result showed the longer wavelength emission was dominant. Only the dual emission for G2 was tunable. However, the detailed mechanistic discussion was not found for why the LE of TPE emission was quenched and how the molecular conformation or intermolecular interactions affect the emission. More insights into these issues should be given.

Re: Response to reviewers' comments for manuscript NCOMMS-17-19577A-Z

We would like to thank all the reviewers for their critical comments and thoughtful suggestions on the manuscript. Enclosed please kindly find the revised manuscript in which we have made our revisions and corrections. The questions, suggestions and comments raised by the reviewers have been addressed in this letter. The detailed summary is as follows:

Reviewer #1:

1) The total species in the chelate solution for G1-G3. The authors did not mention anywhere in the manuscript the association constants in various solvents and how might changing solvent composition affect these constants. I almost have the impression that these rosettes are the sole species in solutions, is that true? But I don't think the authors specified in the manuscript. If this is the case, the authors should provide a simple 2D-fluorescence measurement that indicates independence of emission from excitation for G1-G3 and how solvent composition influences the results. Also, a temperature dependent experiment may also help provided that the rosettes dissociate at higher temperatures.

Figure R1. (a) UV/Vis spectra for the titration experiment as the molar ratio of **L1**/Cd²⁺ is changed from 1:0 to 1:24.71 in a mixed solvent of CHCl₃/MeOH (1/3). The concentration of **L1** is 10 μM. (b) Binding isotherm of **L1** and Cd²⁺. The calculated association constant (K_a) is $2.9 \pm 0.3 \times 10^5 \text{ M}^{-1}$.

Response: In the self-assembly of **G2** and **G3**, hexameric and heptameric rosettes are the sole species in solution. In the case of **G1**, the self-assembly generated a mixture of macrocycles. Due to the multivalent interactions of multi-armed ligands in **G2** and **G3**, we used **G1** as a model system to study the association constants (Fig. S58–61) in different solvents, including THF, CHCl₃/MeOH, DMF and DMSO. For example, as shown in Figure R1, the association constants of **G1** in CHCl₃/MeOH (v/v, 1/3) is $2.9 \pm 0.3 \times 10^5 \text{ M}^{-1}$. The solvent composition does affect the constants, its value would decreased when increasing the polarity of solvents. In THF, the association constants of **G1** is $1.1 \pm 0.2 \times 10^5 \text{ M}^{-1}$, which is close to the value in CHCl₃/MeOH (v/v, 1/3). The association constants of **G1** were decreased by nearly an order of magnitude in DMF ($6.3 \pm 1.4 \times 10^4 \text{ M}^{-1}$) and DMSO ($4.0 \pm 0.5 \times 10^4 \text{ M}^{-1}$).

Figure R2. (a) Multi-wavelength (2D) fluorescence spectrum (excitation step width: 10 nm) of **G2** ($c = 1.0 \mu\text{M}$) in CH_3CN /methanol mixtures with different methanol contents (a) 0 %, (b) 30 %, (c) 60 %, (d) 90 %.

We performed 2D- fluorescence study as suggested by reviewer. For instance, as shown in Fig. R2, the luminous range of 2D-fluorescence of **G2** was consistent with 1D fluorescence. No other light-emitting specie was observed. The 2D-fluorescence results of **G1** (Figs. S67 and S68) and **G3** (Figs. S71 and S72) also show the independence of emission from excitation. As such, the supramolecules did not dissociate at the concentration of fluorescence study.

Figure R3. Temperature dependence of fluorescence emission spectra of **G3** in CH_3CN ($\lambda_{\text{ex}} = 320 \text{ nm}$, $c = 1.0 \mu\text{M}$).

Finally, the temperature dependent study was conducted following reviewer's suggestion. The temperature dependent fluorescence study showed these rosettes are stable at the temperature from -46°C to 50°C . As for **G1** (Fig. S75) and **G3** (Figs. R3 and S77), the emission intensity was increased when decreasing the temperature of the system which is consistent with traditional fluorescence process. As for **G2** (Fig. S76), the local emission of TPE was decreased while the CT emission was increased when

the temperature decreased from 50°C to -46°C , because the intramolecular rotation process is restricted and the electronic communication between TPE and metal becomes dominant (*Chem. Sci.*, 2015, 6, 5347, page 7). Furthermore, variable temperature ^1H NMR spectra of **G2** and **G3** also show no dissociation of supramolecules (Figs. S53-54). Detailed discussion was added in P13 Paragraph 2 in the manuscript.

2) *The interpretation of certain fluorescence results, for instance, the assignment of the shoulder emission at longer wavelength throughout the samples. The authors attribute these peaks to charge-transfer fluorescence while providing no evidence to support it. For example, charge-transfer absorption and fluorescence are typically very sensitive solvent polarity and media rigidity? This should be very easy to prove with just a bit more effort in experiment. Also, the authors explained the enhancement of G1-G3 fluorescence in comparison to L1-L3 in good solvent as inhibition of molecular motions. This is hardly convincing given that the enhancement is not as dramatic. Metal coordination can have a tremendous effect in the electronic transitions of the ligands. I suggest that the author show hard evidence for all the explanations they list in the fluorescence properties section as they did in the characterization part at the beginning of the manuscript, which is nice and clear.*

Figure R4: The emission spectra of **G1** in different solvents ($\lambda_{\text{ex}} = 320 \text{ nm}$, $c = 1.0 \text{ }\mu\text{M}$)

Response: According to the reviewer's suggestion, the fluorescence experiments were performed in different solvents with gradient polarity (Figs. S78–80). For example, as shown in Figure R4, the shape of the emission bands and the emission maxima are sensitive to the solvents used, indicating the emission at longer wavelength could be attributed to MLCT. Detailed discussion was added in P13 Paragraph 2 in the manuscript. TPE-TPY ligands were assembled with Cd(II) through coordination to introduce additional restriction of intramolecular rotation (RIR) and immobilize fluorophores into metallo-supramolecules with rosettes-like scaffolds. Also, by introducing metal coordination, some interesting luminescence properties were brought by red shift of emission bands. Some other information including 2D-fluorescence experiments, VT-fluorescence experiments have been added in SI and discussed in the main text.

3) *The description of the figure legends are not adequately informative (such as the lack of concentration information) and the readers have to fish out some numbers in the main text.*

Response: We have added some detail information to the figure captions. For instance, the concentration information has been added in the Figures 4-7. In addition, we added the excitation of fluorescence experiments in Figures 5-6.

4) *P10 L171 "rear"  "rare"?*

Response: Revised it as "rare".

5) *What is the potential use of white emission in solution since the authors seem to emphasize the concept a lot?*

Response: In the past few years, white organic light-emitting materials, devices, and processes attracted great attention due to their fundamental importance and practical application. However, most light-emitting materials reported so far were based on the combination of multi-components with emission color covering the entire visible range. Single-component white light emitters are expected to exhibit superior performance improved stability, good reproducibility, and simple device fabrication procedure but without phase segregation and color aging compared to these combined emitters (B. Z. Tang *et.al.* *Nat Commun.* 2017, 8, 416). We herein provide an alternative strategy to construct a single-component white light emitter in this study.

5) *I am just curious what the purpose is for choosing Cd(II) over other metals such as Zn(II)?*

Response: 2,2':6',2''-Terpyridine (tpy) ligands are widely used as building blocks in supramolecular and macromolecular chemistry. An extremely interesting aspect of this tridentate ligand is the different binding strengths of tpy and transition metal ions which order as $\text{Ru(II)} > \text{Fe(II)} > \text{Ni(II)} > \text{Zn(II)} > \text{Cd(II)}$ in the

complex of $[M(\text{tpy})_2]^{2+}$. So the reversibility of Zn(II) is not as good as Cd(II). ESI-MS showed many by-products were obtained in addition to target assemblies when ligands self-assembled with Zn(II). By contrast, we are able to obtain discrete assemblies **G2** and **G3** using Cd(II).

6) Measured lifetimes in the ns range do not necessarily mean fluorescence for G1-G3. If the quantum yield is only half a percent in air, then the intrinsic lifetimes are in the sub micron scale, which could involve metal in the triplet excited state.

Figure R5. Time-resolved fluorescence decay profiles of (a) **G1** and (d) **G3** in deaerated CH_3CN ($c = 1.0 \mu\text{M}$). Time-resolved fluorescence decay profiles of **G2** in deaerated CH_3CN ($c = 1.0 \mu\text{M}$) at the two wavelengths (b) at 450 nm, (c) at 580 nm.

Response: In order to elucidate luminescence process, the lifetime measurements have been repeated in degassed solutions but nearly no influence of oxygen was observed. The lifetimes still in the ns scale, which indicates the lifetimes are not sensitive to oxygen (Fig. R5). It indicates the luminescence is fluorescence process. The additional experimental results were depicted in Figure S85.

Reviewer #2:

1) Since the structures of ligands L1-L3 are very complicated, the detailed ^1H NMR assignments should be done more carefully. For example, the two sets of tpy signals should be distinguishable by NOE experiments. In addition, to ensure high purity of ligands, it is suggested the authors provide the whole spectrum of L1-L3 instead of the isotope patterns, or their elemental analysis.

Response: According to reviewer's suggestion, **L1** and **L2** have been further characterized by NOESY (**L1**) (Figs. S13 and S14) and ROESY (**L2**) (Figs. S21 and 22). All proton include tpy signals of L1-L3 have been well assigned carefully characterized by ^1H , 2D COSY, 2D ROESY and NOESY NMR. In addition, we have provided the whole MALDI-TOF spectra of **L1-L3** (Figs. S15, S23 and S31) in the SI as suggested by reviewer.

Figure R6. 2D ROESY NMR (500 MHz, CDCl₃, 300 K) spectrum of ligand **L2**.

Figure R7: MALDI-TOF mass spectrum of ligand **L2**.

2) The complexation of **L1** with Cd(II) ions gave rise to a dynamic library including four macrocycles, which were observed by ESI-MS. Is it possible to have smaller or bigger rings, such as a dimer or a heptamer, in this series? Are those structures concentration-dependent? If yes, the population of the dynamic library should be very different at various concentrations. Therefore, how do the concentration and/or molecular sizes affect the emissive properties?

Response: Indeed, the components of **G1** solution were determined by concentration as shown in the Fig. R8. At lower concentration, we did not observe the formation of heptamer due to entropy-driven self-assembly in ESI-MS. However, no dimer was detected with the concentration from 0.5 mg/mL to 3.0 mg/mL perhaps because of the large ring constraint for dimer. We also conducted variant-concentration fluorescence study of **G1** from 0.25 mg/mL to 3.0 mg/mL. The emission intensity increased with slight blue-shift when increasing the concentration of **G1** (Fig. R9). The additional experimental results have been added in Supporting Information as Fig. S6 in page 27 and Fig. S81 in page 105.

Figure R8. (a) Variable concentration ESI-MS spectra of **G1** in CH₃CN/MeOH (v/v, 3/1) from 0.25 mg/mL to 3.0 mg/mL. (b) Zoom in $m/z = 753, 813, 853, 574, 903$. Macrocycle complexes are named Mn^{x+} , where M designates the repeat unit $\langle tpy-Cd-tpy \rangle_n$, n is the number of repeat units, and x is the number of charges.

Figure R9. The variant-concentration fluorescence of **G1** from 0.25 mg/mL to 3.0 mg/mL ($\lambda_{ex} = 320$ nm)

3) In the assembly of L2 with Cd(II), the resultant complex was assigned to a hexameric structure mainly relied on the ESI-MS evidence. However, there are some minor signals shown in Figure 3a. Can those minor signals be identified? Are they coming from other possible structures? Moreover, the broad ^1H NMR peaks were attributed to slow tumbling motions by the authors. Does the variant-temperature NMR experiment help understand this phenomenon? In the NMR spectrum, some signals are not identified properly, including two small humps at 8.8 and 8.4 ppm and a number of peaks in the range from 2.4 to 4 ppm. Are they generated by other assembled structures? If so, "quantitative yield" would be not true.

Response: Because of the labile interaction of metal-ligand coordination, we had to use very mild electrospray ionization condition to maintain the integrity of supramolecules, including low desolvation temperature and low ESI cone voltage. Therefore, supramolecules were readily to form solvent adducts or salts adducts in ESI-MS. Fujita and other groups reported similar results, particularly in the characterization of large metallo-supramolecules using cold-spray ionization mass spectrometry (*J. Mass Spectrom.* 2003, 38, 473; *Nat. Chem.* 2012, 4, 330; *Angew. Chem. Int. Ed* 2016, 55, 445). Furthermore, ESI-MS still cannot totally prevent the dissociation of metallo-supramolecules although it is a soft ionization technique. The dissociation could induce the formation of the minor peaks in Figure 3a.

Figure R10: Variable temperature ^1H NMR spectra (500 MHz) of **G2** in CD_3CN (from 293 K to 333 K)

We conducted variant-temperature NMR experiments (from 293K to 333K) according to reviewer's suggestion. As shown in Fig. R10, the peaks in ^1H NMR of **G2** were getting much shaper when increasing temperature. It indicates that the broad ^1H NMR peaks were attributed to slow tumbling motions of rigid skeleton of rosettes. Some signals of impurities at 8.8 and 8.4 ppm and from 2.4 to 4 ppm in the NMR spectrum of **G2** come from introducing solvents incautiously. We have re-performed all NMR (^1H NMR, ^{13}C NMR, COSY, DOSY and NOESY) of **G2** again which are pure without impurities. All the spectra were updated in manuscript and SI. Also we have recalculated and revised the yield of **G2** and the "quantitative yield" has been removed from text following your suggestion. The additional experimental results have been added in Supporting Information.

4) In the case of G3, again the broad ^1H NMR peaks and only a few signals in the ^{13}C NMR spectrum were observed. There are even more noisy signals shown in the ESI-MS spectrum. On the basis of those data, the structure of the uncommon heptameric macrocycle cannot be established unambiguously. At least, the whole ESI-MS spectrum of G3 should be provided and carefully assigned to exclude the possibility of forming other species.

Figure R11: Variable temperature ^1H NMR spectra (500 MHz) of **G3** in CD_3CN (from 293 K to 333 K)

Response: We have carried out variant-temperature NMR experiments of **G3** (from 293K to 333K). As shown in Fig. R11, the peaks in ^1H NMR of **G3** were getting much shaper when increasing temperature as **G2**. It indicates that the broad ^1H NMR peaks were attributed to slow tumbling motions of rigid skeleton.

Figure R12. ^{13}C DEPT 45° NMR (100 MHz, CD_3CN , 300 K) spectrum of **G3**.

It's a challenge to achieve high quality of ^{13}C NMR spectra of **G2** and **G3** with high molecular weight (14,498 Da and 27,599 Da, respectively) due to the poor solubility and sensitivity. Generally, ^{13}C DEPT NMR has better sensitivity than normal ^{13}C NMR. To obtain better quality of ^{13}C NMR spectra of **G2** and **G3**, we have performed ^{13}C DEPT 45° NMR of **G2** and **G3**. For example, as shown in Fig. R12, all aliphatic carbon signals were collected and several aromatic carbon signals of **G3** were also observed.

As mentioned in the ESI-MS of **G2**, we typically used very mild ionization condition for metallo-supramolecules to minimize the fragmentation. However, due to the high molecular weight of **G3**, we had to increase the ionization voltage to improve the ionization efficiency and increase the sensitivity. Such operation created more fragments in ESI-MS spectrum. The whole ESI-MS spectrum of **G3** (Fig. S2) has been provided in SI and the minor signals were assigned carefully. In addition, the traveling-wave ion mobility mass spectrometry (TWIM-MS) displays a narrowly distributed band of signals suggesting the formation of a rigid and discrete assembly. The additional experimental results have been added in Supporting Information.

5) Although the tube-like morphologies were observed in the TEM images, the nanostructures cannot be directly correlated to the proposed molecular packing without any electron or x-ray diffraction evidence.

Figure R13. Powder XRD result of **G2** nanotube.

Response: We conducted electron diffraction for the tubular nanostructures but did not get satisfactory diffraction signals perhaps due to the small size of nanostructures. We then used powder x-ray diffraction to characterize **G2**, which gave broad signals as shown in Fig. R13. The highest peak is corresponding to 4 Å spacing. Unfortunately, we were unable to get more conclusive evidence from powder x-ray diffraction. Compared to the study Aida and coworkers (*Science* 2014, 344, 499), we speculate that the alignment of tubular nanostructures to form more uniform sample for either electron or x-ray diffraction is the major issue we need to solve in the ongoing study.

6) In the manuscript, the emission at shorter wavelength was attributed to the LE state of TPE and the one at longer wavelength was assigned to ICT from TPE to tpy units. The authors should provide experimental evidence to say so. In addition, since the TPE in **G1** has higher flexibility, it would be expected to observe the emission at around 430 nm from TPE at higher solvent polarity due to AIE, but the result showed the longer wavelength emission was dominant. Only the dual emission for **G2** was tunable. However, the detailed mechanistic discussion was not found for why the LE of TPE emission was quenched and how the molecular conformation or intermolecular interactions affect the emission. More insights into these issues should be given.

Response: To better understand the fluorescence process, we have done 2D-fluorescence, variant-concentration fluorescence experiments as well as fluorescence in different solvents with various polarity. As mentioned above, the emission maxima and the shape of the emission bands of **G1-G3** rely strongly on the polarity of solvents, suggesting that the emission at longer wavelength should be derived from ICT.

Figure R14. (a) Fluorescence spectra of **G1** in CH₃CN/H₂O mixtures with different water contents ($\lambda_{\text{ex}} = 320$ nm, $c = 1.0$ μM), and (b) Zoom in 350 – 500 nm.

Actually, the LE emissions of **G1** (Fig. R14) and **G3** (Figure 6a) do exist, although the intensity is low, due to the flexibility of the TPE backbone (**G1**) or relatively weaker intensity compared with ICT emission (**G3**). Furthermore, as shown in Figs. R15-R17, the overlapped emission spectra and the absorption spectra of **G1-G3** are prone to the energy transfer (ET) process, and quench the LE emission. Benefited from the relative rigid skeleton (compared with **G1**) and medium ICT effect, dual emission was found in **G2** system

and tunable. In addition, the strongest ICT effect of **G3** among three rosettes leads to the longer wavelength emission dominant.

Figure 15. Normalized absorption spectrum and fluorescence spectrum of **G1** in $\text{CH}_3\text{CN}/\text{H}_2\text{O}$ mixtures with 10% water contents ($\lambda_{\text{ex}} = 320 \text{ nm}$, $c = 1.0 \mu\text{M}$).

Figure 16. Normalized absorption spectrum and fluorescence spectrum of **G2** in $\text{CH}_3\text{CN}/\text{H}_2\text{O}$ mixtures with 0% water contents ($\lambda_{\text{ex}} = 320 \text{ nm}$, $c = 1.0 \mu\text{M}$).

Figure 17. Normalized absorption spectrum and fluorescence spectrum of **G3** in $\text{CH}_3\text{CN}/\text{MeOH}$ mixtures with 90% methanol contents ($\lambda_{\text{ex}} = 320 \text{ nm}$, $c = 1.0 \mu\text{M}$).

I hope that the revision will satisfy our reviewers' requirements. Again, I would like to thank the reviewers for their thoughtful and careful comments and appreciate very much for your kindly handling this manuscript.

Reviewers' comments:

Reviewer #1 (Remarks to the Author):

The authors did a great job addressing my previous questions by conducting more thorough experiments. The responses are professional and adequately sufficient; I would recommend for its timely publication in Nature Communications.

Reviewer #2 (Remarks to the Author):

The structural characterization of ligands and complexes has been improved, but there are a few comments for the authors to consider.

(1) Since a better ¹H NMR spectrum of G2 was obtained, Figure 2a should be replaced by the better one.

(2) The whole ESI-MS spectrum of G3 with the identified fragment signals (Figure S2) should be presented in Figure 3c instead. In addition, tandem MS experiments may help clarify the source of fragments, which the authors attributed to the harsh ionization conditions.

(3) Molecular simulation may provide insights into why a heptamer was formed instead of a hexamer in the complexation reaction of L3 by comparing their energy states.

(4) The association constants of G1 in various solvents were calculated by UV/vis titration experiments. However, there are many species generated from the complexation reaction between L1 and Cd(II). In addition, in the presence of excess metal ions (L:M = 1:24.5), the assembled structure may further changed to a bisterpyridine-metal (1:2) adduct. The authors did not specify which equilibrium they looked into and which method and model they used for the curve fitting. The coefficients of determination were pretty bad in Figures S59 and S60. Since this is a complicated dynamic system, the calculation should be done more carefully.

I support this manuscript to be published in Nature Communications after the mentioned issues are properly addressed.

Re: Response to reviewers' comments for manuscript NCOMMS-17-19577B

To whom it may concern:

We would like to thank all the reviewers for their critical comments and thoughtful suggestions on the manuscript. Enclosed please kindly find the revised manuscript in which we have made our revisions and corrections. The questions, suggestions and comments raised by the reviewer 2 have been addressed in this letter. The detailed summary is as follows:

Reviewer #2:

(1) Since a better ^1H NMR spectrum of G2 was obtained, Figure 2a should be replaced by the better one.

Response: The ^1H NMR spectrum of G2 in Fig.2a was updated.

(2) The whole ESI-MS spectrum of G3 with the identified fragment signals (Figure S2) should be presented in Figure 3c instead. In addition, tandem MS experiments may help clarify the source of fragments, which the authors attributed to the harsh ionization conditions.

Figure R1: ESI-MS of G3 with different ESI cone voltage (a) 4 kV, (b) 3 kV, (c) 2 kV.

Response: The Figure 3c was updated with labeled fragments for the whole ESI-MS spectrum of G3. We have performed tandem MS using collisionally-induced dissociation (CID) following Reviewer's suggestion. In CID, however, supramolecules were prone to losing different number of neutral PF₅ instead of forming large fragments (*Angew. Chem. Int. Ed.*, 2010, 49, 6539). To clarify the source of fragment, we further performed ESI-MS experiments of G3 with different ESI cone voltages. As shown in Figure R1, the signals of fragments increased when increasing the ESI cone voltage. Trace fragments could be observed when the ionization voltage is 2 kV (Figure R1c). The fragments were further increased especially at low m/z region when cone voltage up to 4 kV (Figure R1a). The further discussion on the fragment signals of G3 was added in P7 Paragraph 2 in the manuscript and the additional experimental results was added in Figure S3.

(3) Molecular simulation may provide insights into why a heptamer was formed instead of a hexamer in the complexation reaction of L3 by comparing their energy states.

Figure R2. The modeling structures of heptamer and hexamer assembled by **L3** and Cd^{2+} . (a) topview, (b) sideview of heptamer and (c) topview (d) sideview of hexamer.

Response: We have conducted molecular simulation for hexamer and heptamer assembled by **L3** and Cd^{2+} . As shown in Figure R2, the optimum structure of hexamer was highly distorted. In contrast, the structure of heptamer exhibited more extended skeleton with lower distortion. The torsion energy of heptamer (499.27 kcal/mol) is lower than hexamer (567.91 kcal/mol). Therefore, the self-assembly preferred the formation of heptamer rather than hexamer. Detailed discussion was added in P7 Paragraph 2 in the manuscript.

(4) The association constants of **G1** in various solvents were calculated by UV/vis titration experiments. However, there are many species generated from the complexation reaction between **L1** and $\text{Cd}(\text{II})$. In addition, in the presence of excess metal ions ($L:M = 1:24.5$), the assembled structure may further changed to a bisterpyridine-metal (1:2) adduct. The authors did not specify which equilibrium they looked into and which method and model they used for the curve fitting. The coefficients of determination were pretty bad in Figures S59 and S60. Since this is a complicated dynamic system, the calculation should be done more carefully. I support this manuscript to be published in Nature Communications after the mentioned issues are properly addressed.

Response: Thank you for Reviewer's suggestion. The bisterpyridine-metal (1:2) adduct was not taken into account when added excess amount of Cd^{2+} in our first response letter.

Scheme R1. (I) Less than 1 equiv. $\text{Cd}(\text{NO}_3)_2$; (II) more than 1.0 equiv. $\text{Cd}(\text{NO}_3)_2$ binding with **L1**

We recalculated the association constants of **G1** on 1:1 binding model (Scheme R1I) by Benesi-Hildebrand method (Equation 1) according to the reported literature (*Spectrochim. Acta A*, 2012, 90, 40–44). For example, as shown in Figure R3, the association constants of **G1** in THF is $1.24 \times 10^5 \text{ M}^{-1}$, which is close to the value ($1.1 \times 10^5 \text{ M}^{-1}$) calculated before. The measured absorbance $[1/(A - A_0)]$ varied as a function of $1/[\text{Cd}^{2+}]$ in a linear relationship ($R = 0.997$), which is consistent with 1:1 complex formation. Similarly, the association constants

in CHCl₃/MeOH (v/v, 1/3), DMF and DMSO were all recalculated with satisfactory coefficients by this method. Note that in Benesi–Hildebrand plot, we only used the absorbance with L:M ratio slightly larger than 1:1 to calculate K_a. The association constants in various solvents have been updated in the manuscript and Supporting Information as Figure S60-63 in page S82-85.

The association constant (K_a) of **L1** with Cd²⁺ was determined using the Benesi–Hildebrand equation as follows:

$$\frac{1}{A - A_0} = \frac{1}{K_a(A_{max} - A_0)[Cd^{2+}]} + \frac{1}{A_{max} - A_0} \quad (1)$$

where A and A₀ represent the absorbance of **L1** in the presence and absence of Cd²⁺, respectively, A_{max} is the saturated absorbance of **L** after the addition of excess amount of Cd²⁺; [Cd²⁺] is the concentration of Cd²⁺ ion added. K_a is the association constant.

Figure R3. (a) UV/Vis spectra for the titration experiment as the molar ratio of **L1**/Cd²⁺ is changed from 1:0 to 1:1.33 in a solvent of THF. The concentration of **L1** is 10 μM. (b) K_a (1.24 × 10⁵ M⁻¹, R = 0.997) was calculated using Benesi–Hildebrand plot of **L1** with Cd²⁺. The measured absorbance 1/(A-A₀) was observed at 343 nm as a function of the 1/[Cd²⁺].

I hope that the revision will satisfy our reviewers' requirements. Again, I would like to thank the reviewers for their thoughtful and careful comments and appreciate very much for your kindly handling this manuscript.

REVIEWERS' COMMENTS:

Reviewer #2 (Remarks to the Author):

The Benesi-Hildebrand method the authors used for calculating association constants is a specific approach for one-to-one complex systems ($M + L \rightarrow ML$). However, the self-assembly of G1 gave rise to a mixture of macrocycles including trimer, tetramer, pentamer and hexamer. Each species has its own equilibrium constant. Hence, the B-H method cannot be applied to this $[M_a L_a]$ system (please see Inorg. Chem. 1994, 33, 972-981 for details). Taking the trimer formation as an example, the equilibrium equation is $3M + 3L \rightarrow M_3L_3$ where the association constant should be expressed as $K_a = \frac{[M_3L_3]}{[M]^3[L]^3}$. The unit for the K_a is M^{-5} . The incorrect calculation shown in the manuscript should be properly fixed before publication.

Reviewer #2 (Comments to the Author):

*The Benesi-Hildebrand method the authors used for calculating association constants is a specific approach for one-to-one complex systems ($M + L \rightarrow ML$). However, the self-assembly of **G1** gave rise to a mixture of macrocycles including trimer, tetramer, pentamer and hexamer. Each species has its own equilibrium constant. Hence, the B-H method cannot be applied to this $[M_nL_n]$ system (please see *Inorg. Chem.* 1994, 33, 972-981 for details). Taking the trimer formation as an example, the equilibrium equation is $3M + 3L \rightarrow M_3L_3$ where the association constant should be expressed as $K_a = [M_3L_3]/[M]^3[L]^3$. The unit for the K_a is M^{-5} . The incorrect calculation shown in the manuscript should be properly fixed before publication.*

Response: Thank you for Reviewer's thoughtful suggestion. According to Reviewer #1's request for the stability of complexes, we investigated association constants in different solvent. We agreed with Reivewer #1's concern about the multiple assemblies in **G1** system. To overcome this problem, we synthesized a new monodentate compound **16** as a model system to determine the binding strength of terpyridine with Cd^{2+} . We performed the titration experiments in $CHCl_3/MeOH$ (1/2), THF, and DMF in consideration of solubility and polarity. Association constants for all titrations were calculated using Bindfit v0.5 (Open Data Fit, 2016, <http://app.supramolecular.org/bindfit/>; *Nature Chemistry*, 2017, **9**, 903–908) based on 2:1 binding model (Scheme R1). The equation used for these analyses is available in the review by Thordarson (*Chem. Soc. Rev.* 2011, **40**, 1305-1323). For example, as shown in Figure R1, the calculated association constant in $CHCl_3/MeOH$ (1/2) is $4.17 \times 10^{10} M^{-2}$. There is no significant difference for association constants in different solvents. It should be noted that the stability of multi-armed ligands **L1-L3** binding with Cd^{2+} should higher than monoterpyridine compound **16**.

1H , ^{13}C NMR and high resolution ESI-TOF mass spectrometry information of compound **16** were provided in Page S19. The further discussion on the determination of association constants of compound **16** binding with Cd^{2+} was added in Page S83 and the additional experimental results were added in Supplementary Figures 65-67.

Scheme R1. Equilibrium for binding Cd²⁺ to compound **16**, where K_1 , K_2 are the first and the second binding association constants, respectively, K is the overall association constant .

Figure R1. (a) UV/Vis spectra for the titration experiment, molar ratio of compound **16**/Cd²⁺ is changed from 1:0 to 1:0.64 in a mixed solvent of CHCl₃/MeOH (1/2). The concentration of compound **16** is 10 μM; (b) Binding isotherms (2:1 model) fitted to the absorbance shift vs. the equivalents of Cd²⁺ added to determine the association constant (top), the calculated association constant K is $4.17 \times 10^{10} \text{ M}^{-2}$; and the residual plot from the fit (bottom).